# RoBERTa: A Robustly Optimized BERT Pretraining Approach

## Abstract

Language model pretraining has led to significant performance gains but careful comparison between different approaches is challenging. Training is computationally expensive, often done on private datasets of different sizes, and, as we show, hyperparameter choices have significant impact on the final results. We present a replication study of BERT pretraining (Devlin et al., 2019) that carefully measures the impact of many key hyperparameters and training data size. We find that BERT was significantly undertrained, and can match or exceed the performance of every model published after it. Our best model achieves state-of-the-art results on GLUE, RACE, SQuAD, SuperGLUE and XNLI. These results highlight the importance of previously overlooked design choices, and raise questions about the source of recently reported improvements. We release our models and code.[1]

## 1 Introduction

Self-training methods such as ELMo (Peters et al., 2018), GPT (Radford et al., 2018), BERT (Devlin et al., 2019), XLM (Lample & Conneau, 2019), and XLNet (Yang et al., 2019) have brought significant performance gains, but it can be challenging to determine which aspects of the methods contribute the most. Training is computationally expensive, limiting the amount of tuning that can be done, and modeling advances are often conflated with changes in data size or composition.

We present a replication study of BERT pretraining (Devlin et al., 2019), which includes a careful evaluation of the effects of hyperparameter tuning and training set size. We find that BERT was significantly undertrained and propose an improved training recipe, which we call RoBERTa, that can match or exceed the performance of all of the post-BERT methods. Our modifications are simple, they include: (1) training the model longer, with bigger batches, over more data; (2) removing the next sentence prediction objective; (3) training on longer sequences; and (4) dynamically changing the masking pattern applied to the training data. We also collect a large new dataset (CC-News) of comparable size to other privately used datasets, to better control for training set size effects.

When controlling for training data, our improved training procedure improves upon the published BERT results on the GLUE (Wang et al., 2019b) and SQuAD (Rajpurkar et al., 2016) benchmarks. When trained for longer over additional data, our model achieves a score of 88.5 on the public GLUE leaderboard, matching the 88.4 reported by Yang et al. (2019). Our model establishes a new state-of-the-art on 4/9 of the GLUE tasks, as well as RACE (Lai et al., 2017), SuperGLUE (Wang et al., 2019a), and XNLI (Conneau et al., 2018), and matches the state-of-the-art on SQuAD. Overall, we re-establish that BERT's masked language model training objective is competitive with recently proposed alternatives such as perturbed autoregressive language modeling (Yang et al., 2019).[2]

In summary, the contributions of this paper are: (1) We present a set of important BERT design choices and training strategies and introduce alternatives that lead to better downstream task performance; (2) We use a novel dataset, CC-News, and confirm that using more data for pretraining further improves performance on downstream tasks; (3) Our training improvements show that masked language model pretraining, under the right design choices, is competitive with all other recently published methods. We release our model, pretraining and fine-tuning code.

---

[1]Our models and code are available at: `anonymous URL`.

[2]These other methods could possibly improve with more tuning as well; we leave this to future work.

## 2 BACKGROUND

**Setup:** BERT (Devlin et al., 2019) takes as input a concatenation of two segments (sequences of tokens), $x_1, \ldots, x_N$ and $y_1, \ldots, y_M$. Segments usually consist of more than one natural sentence. The two segments are presented as a single input sequence to BERT with special tokens delimiting them: $[CLS], x_1, \ldots, x_N, [SEP], y_1, \ldots, y_M, [EOS]$. $M$ and $N$ are constrained such that $M+N < T$, where $T$ is a parameter that controls the maximum sequence length during training.

**Architecture:** BERT uses the now ubiquitous transformer architecture (Vaswani et al., 2017), which we will not review in detail. We use a transformer architecture with $L$ layers. Each block has $A$ self-attention heads and hidden dimension $H$.

**Training Objectives:** BERT uses two pretraining objectives: masked language modeling and next sentence prediction. For the *Masked Language Model (MLM)* objective, BERT is trained via a cross-entropy loss to predict 15% of the input tokens, selected at random. To prevent the model from cheating, 80% of these selected tokens are replaced by a special $[MASK]$ symbol in the input, 10% are replaced by a random token from the vocabulary, and 10% are left unchanged.

*Next Sentence Prediction (NSP)* is a binary classification loss for predicting whether two segments follow each other in the original text. Positive examples are created by taking consecutive sentences from the text corpus. Negative examples are created by pairing segments from different documents. Positive and negative examples are sampled with equal probability.

**Optimization:** BERT is optimized with AdamW (Kingma & Ba, 2015) using the following parameters: $\beta_1 = 0.9$, $\beta_2 = 0.999$, $\epsilon = $ 1e-6 and decoupled weight decay of 0.01 (Loshchilov & Hutter, 2019). The learning rate is warmed up over the first 10,000 steps to a peak value of 1e-4, and then linearly decayed. BERT trains with a dropout of 0.1 on all layers and attention weights, and a GELU activation function (Hendrycks & Gimpel, 2016). Models are pretrained for $S = $ 1,000,000 updates, with mini-batches containing $B = 256$ sequences of maximum length $T = 512$ tokens.

**Data:** BERT is trained on a combination of BOOKCORPUS (Zhu et al., 2015) plus English WIKIPEDIA, which totals 16GB of uncompressed text.[3]

## 3 EXPERIMENTAL SETUP

### 3.1 IMPLEMENTATION

We reimplement BERT in FAIRSEQ (Ott et al., 2019). We primarily follow the original BERT optimization hyperparameters, given in Section 2, except for the peak learning rate and number of warmup steps, which are tuned separately for each setting. We found training to be very sensitive to the Adam epsilon term, and in some cases we obtained better performance or improved stability after tuning it. We also set $\beta_2 = 0.98$ to improve stability when training with large batch sizes.

We pretrain with sequences of at most $T = 512$ tokens. Unlike Devlin et al. (2019), we do not randomly inject short sequences, and we do not train with a reduced sequence length for the first 90% of updates. We train only with full-length sequences.

We train with mixed precision floating point arithmetic on DGX-1 machines, each with $8 \times 32$GB Nvidia V100 GPUs interconnected by Infiniband (Micikevicius et al., 2018).

### 3.2 DATA

BERT-style pretraining crucially relies on large quantities of text. Baevski et al. (2019) demonstrate that increasing data size can result in improved end-task performance. Several efforts have trained on datasets larger and more diverse than the original BERT (Radford et al., 2019; Yang et al., 2019; Zellers et al., 2019). Unfortunately, not all of the additional datasets can be publicly released. For

---

[3]Yang et al. (2019) use the same dataset but report having only 13GB of text after data cleaning. This is most likely due to subtle differences in the underlying data collection or preprocessing.

our study, we focus on gathering as much data as possible for experimentation, allowing us to match the overall quality and quantity of data as appropriate for each comparison.

We consider five English-language corpora of varying sizes and domains, totaling over 160GB of uncompressed text: (1&2) BOOKCORPUS (Zhu et al., 2015) plus English WIKIPEDIA, which is the original data used to train BERT (16GB); (3) CC-NEWS, which we collect from the English portion of the CommonCrawl News dataset (Nagel, 2016), containing 63 million English news articles crawled between September 2016 and February 2019 (76GB after filtering);[4] (4) OPEN-WEBTEXT (Gokaslan & Cohen, 2019), an open-source recreation of the WebText corpus described in Radford et al. (2019), containing web content extracted from URLs shared on Reddit with at least three upvotes (38GB);[5] (5) STORIES, a dataset introduced in Trinh & Le (2018) containing a subset of CommonCrawl data filtered to match the story-like style of Winograd schemas (31GB).

## 3.3 EVALUATION

Following previous work, we evaluate our pretrained models by finetuning on downstream tasks:

- **GLUE:** The General Language Understanding Evaluation (GLUE) benchmark (Wang et al., 2019b) is a collection of 9 datasets for evaluating natural language understanding systems. Tasks are framed as either single-sentence classification or sentence-pair classification tasks. The GLUE organizers provide training and development data splits as well as a submission server and leaderboard that allows participants to evaluate and compare their systems on private held-out test data.
- **SQuAD:** The Stanford Question Answering Dataset (SQuAD) provides a paragraph of context and a question. The task is to answer the question with a span extracted from the context. We evaluate on SQuAD V1.1 and V2.0 (Rajpurkar et al., 2016; 2018). In V1.1 the context always contains an answer, while in V2.0 some questions are not answered in the provided context.
- **RACE:** ReAding Comprehension from Examinations (RACE) (Lai et al., 2017) is a large-scale reading comprehension dataset collected from English examinations in China. The task is to choose among four possible answers to a given question, using a given passage of text as context.
- **Additional Benchmarks:** In the Appendix we present additional results for SuperGLUE (Wang et al., 2019a) and XNLI (Conneau et al., 2018).

## 4 TRAINING PROCEDURE ANALYSIS

This section explores and quantifies which choices are important for successfully pretraining BERT models. We keep the model architecture fixed.[6] Specifically, we begin by training BERT models with the same configuration as BERT$_{\text{BASE}}$ ($L = 12$, $H = 768$, $A = 12$, 110M params).

### 4.1 STATIC VS. DYNAMIC MASKING

As discussed in Section 2, BERT relies on predicting randomly masked tokens. The original BERT implementation performed masking once during data preprocessing, resulting in a single *static* mask. To avoid repeating the same masks at every epoch, training data was duplicated 10 times prior to preprocessing, so that each training sequence was seen with the same mask only four times over the course of 40 training epochs. We instead train with *dynamic masking*, where we generate the masking pattern on-the-fly each time we input a sequence to the model. This becomes crucial when pretraining for more steps or with larger datasets, and additionally performs marginally better than static masking on some downstream tasks (see Appendix A).

### 4.2 MODEL INPUT FORMAT AND NEXT SENTENCE PREDICTION

In the original BERT pretraining procedure, the model observes two concatenated document segments and is trained via an auxiliary Next Sentence Prediction (NSP) loss to predict whether these segments were sampled contiguously from the same document or from distinct documents.

---

[4]We use `news-please` (Hamborg et al., 2017) to collect and extract CC-NEWS. CC-NEWS is similar to the REALNEWS dataset described in Zellers et al. (2019).

[5]The authors and their affiliated institutions are not affiliated with the creation of the OpenWebText dataset.

[6]Studying architectural changes, including larger architectures, is an important area for future work.

| Model | SQuAD 1.1/2.0 | MNLI-m | SST-2 | RACE |
|---|---|---|---|---|
| *Our reimplementation (with NSP loss):* | | | | |
| SEGMENT-PAIR | 90.4/78.7 | 84.0 | 92.9 | 64.2 |
| SENTENCE-PAIR | 88.7/76.2 | 82.9 | 92.1 | 63.0 |
| *Our reimplementation (without NSP loss):* | | | | |
| FULL-SENTENCES | 90.4/79.1 | 84.7 | 92.5 | 64.8 |
| DOC-SENTENCES | 90.6/79.7 | 84.7 | 92.7 | 65.6 |
| BERT$_{\text{BASE}}$ | 88.5/76.3 | 84.3 | 92.8 | 64.3 |
| XLNet$_{\text{BASE}}$ (K = 7) | –/81.3 | 85.8 | 92.7 | 66.1 |
| XLNet$_{\text{BASE}}$ (K = 6) | –/81.0 | 85.6 | 93.4 | 66.7 |

Table 1: Development set results for base models pretrained over BOOKCORPUS and WIKIPEDIA. All models are trained for 1M steps with a batch size of 256 sequences. We report F1 for SQuAD and accuracy for MNLI-m, SST-2 and RACE. Reported results are medians over five random initializations (seeds). Results for BERT$_{\text{BASE}}$ and XLNet$_{\text{BASE}}$ are from Yang et al. (2019).

The NSP objective was designed to improve performance on downstream tasks, such as Natural Language Inference (Bowman et al., 2015), which require predicting relationships between pairs of sentences. Devlin et al. (2019) observe that removing NSP hurts performance, with significant performance degradation on QNLI, MNLI, and SQuAD 1.1. However, recent work has questioned the necessity of the NSP loss (Lample & Conneau, 2019; Yang et al., 2019; Joshi et al., 2019).

To better understand this discrepancy, we compare several alternative training formats:

- SEGMENT-PAIR+NSP: This follows the original input format used in BERT (Devlin et al., 2019), with the NSP loss. Each input has a pair of segments, which can each contain multiple natural sentences, but the total combined length must be less than 512 tokens.

- SENTENCE-PAIR+NSP: Each input contains a pair of natural *sentences*, either sampled from a contiguous portion of one document or from separate documents. Since these inputs are significantly shorter than 512 tokens, we increase the batch size so that the total number of tokens remains similar to SEGMENT-PAIR+NSP. We retain the NSP loss.

- FULL-SENTENCES: Each input is packed with full sentences sampled contiguously from one or more documents, such that the total length is at most 512 tokens. Inputs may cross document boundaries. When we reach the end of one document, we begin sampling sentences from the next document and add an extra separator token between documents. We remove the NSP loss.

- DOC-SENTENCES: Inputs are constructed similarly to FULL-SENTENCES, except that they may not cross document boundaries. Inputs sampled near the end of a document may be shorter than 512 tokens, so we dynamically increase the batch size in these cases to achieve a similar number of total tokens as FULL-SENTENCES. We remove the NSP loss.

**Results** Table 1 shows results for the four different settings. We first compare the original SEGMENT-PAIR input format from Devlin et al. (2019) to the SENTENCE-PAIR format; both formats retain the NSP loss, but the latter uses single sentences. We find that **using individual sentences hurts performance on downstream tasks**, which we hypothesize is because the model is not able to learn long-range dependencies.

We next compare training without the NSP loss and training with blocks of text from a single document (DOC-SENTENCES). We find that this setting outperforms the originally published BERT$_{\text{BASE}}$ results and that **removing the NSP loss matches or slightly improves downstream task performance**, in contrast to Devlin et al. (2019). It is possible that the original BERT implementation may only have removed the loss term while still retaining the SEGMENT-PAIR input format.

Finally we find that restricting sequences to come from a single document (DOC-SENTENCES) performs slightly better than packing sequences from multiple documents (FULL-SENTENCES). However, because the DOC-SENTENCES format results in variable batch sizes, we use FULL-SENTENCES in the remainder of our experiments for easier comparison with related work.

| batch size | learning rate | epochs | steps | perplexity | MNLI-m | SST-2 |
|---|---|---|---|---|---|---|
| 256 | 1e-4 | 32 | 1M | 3.99 | 84.7 | 92.5 |
| 2K | 7e-4 | 32 | 125K | 3.68 | 85.2 | 93.1 |
| | | 64 | 250K | 3.59 | 85.3 | **94.1** |
| | | 128 | 500K | 3.51 | 85.4 | 93.5 |
| 8K | 1e-3 | 32 | 31K | 3.77 | 84.4 | 93.2 |
| | | 64 | 63K | 3.60 | 85.3 | 93.5 |
| | | 128 | 125K | **3.50** | **85.8** | **94.1** |

Table 2: Perplexity on held-out validation data and dev set accuracy on MNLI-m and SST-2 for various batch sizes (# sequences) as we vary the number of passes (epochs) through the BOOKS + WIKI data. Reported results are medians over five random initializations (seeds). The learning rate is tuned for each batch size. All results are for BERT$_{\text{BASE}}$ with FULL-SENTENCE inputs.

## 4.3 TRAINING WITH LARGE BATCHES

Past work in neural machine translation has shown that training with large mini-batches can improve optimization speed and end-task performance when the learning rate is tuned appropriately (Ott et al., 2018). Large batches are also easily parallelized via data parallel training.[7]

Table 2 shows the masked LM perplexity and end-task accuracy for BERT$_{\text{BASE}}$ as we increase the batch size, while tuning the learning rate. Devlin et al. (2019) originally trained BERT$_{\text{BASE}}$ for 1M steps with a batch size of 256 sequences; however a batch size of 2K sequences performs better, even controlling for the number of epochs, suggesting that **the original BERT batch size was too small**. We also observe that training with extremely large batches (8K) becomes more efficient as we train for more epochs.[8] In the remainder of our experiments we train with batches of 8K sequences.

## 4.4 TEXT ENCODING

Byte-Pair Encoding (BPE) (Sennrich et al., 2016) is a hybrid between character- and word-level modeling based on subwords units. BPE vocabulary sizes typically range from 10K-100K subword units; however, unicode characters can account for a sizeable portion of this vocabulary when modeling large and diverse corpora, such as the ones considered in this work.

The original BERT implementation (Devlin et al., 2019) used a character-level BPE vocabulary of size 30K. We instead adopt the larger byte-level BPE vocabulary of size 50K introduced in Radford et al. (2019), which uses *bytes* rather than unicode characters as the base subword units and can therefore encode any input text without introducing "unknown" tokens. This adds approximately 15M and 20M extra parameters for BERT$_{\text{BASE}}$ and BERT$_{\text{LARGE}}$, respectively.

Early experiments revealed only minor differences between these encodings, with the byte-level BPE achieving slightly worse end-task performance on some tasks. Nevertheless, we believe the advantages of a universal encoding scheme outweighs the minor degredation in performance and use this encoding in the remainder of our experiments.

## 5 ROBERTA

In the previous section we propose modifications to the BERT pretraining procedure that improve end-task performance. We now aggregate these improvements and evaluate their combined impact. We call this configuration **RoBERTa** for **R**obustly **o**ptimized **BERT** **a**pproach. Specifically, RoBERTa is trained with dynamic masking (Section 4.1), FULL-SENTENCES without NSP loss (Section 4.2), large mini-batches (Section 4.3) and a larger byte-level BPE (Section 4.4).

---

[7]Even without large scale parallel hardware, large batch training can improve training efficiency through *gradient accumulation* – i.e., accumulating gradients from multiple mini-batches before each optimization step.

[8]You et al. (2019) train BERT with even larger batch sizes, up to 32K sequences. We leave further exploration of the limits of large batch training to future work.

| Model | data | batch size | steps | SQuAD (v1.1/2.0) | MNLI-m | SST-2 |
|---|---|---|---|---|---|---|
| RoBERTa | | | | | | |
|    with BOOKS + WIKI | 16GB | 8K | 100K | 93.6/87.3 | 89.0 | 95.3 |
|    + additional data (§3.2) | 160GB | 8K | 100K | 94.0/87.7 | 89.3 | 95.6 |
|    + pretrain longer | 160GB | 8K | 300K | 94.4/88.7 | 90.0 | 96.1 |
|    + pretrain even longer | 160GB | 8K | 500K | **94.6/89.4** | **90.2** | **96.4** |
| BERT$_{\text{LARGE}}$ | | | | | | |
|    with BOOKS + WIKI | 13GB | 256 | 1M | 90.9/81.8 | 86.6 | 93.7 |
| XLNet$_{\text{LARGE}}$ | | | | | | |
|    with BOOKS + WIKI | 13GB | 256 | 1M | 94.0/87.8 | 88.4 | 94.4 |
|    + additional data | 126GB | 2K | 500K | 94.5/88.8 | 89.8 | 95.6 |

Table 3: Development set results for RoBERTa as we pretrain over more data (16GB → 160GB of text) and pretrain for longer (100K → 300K → 500K steps). Each row accumulates improvements from the rows above. RoBERTa matches the architecture and training objective of BERT$_{\text{LARGE}}$. Results for BERT$_{\text{LARGE}}$ and XLNet$_{\text{LARGE}}$ are from Devlin et al. (2019) and Yang et al. (2019), respectively. Complete results on all GLUE tasks can be found in Appendix C.

Additionally, we investigate two other important factors that have been under-emphasized in previous work: (1) the data used for pretraining, and (2) the number of training passes through the data. For example, XLNet (Yang et al., 2019) was pretrained using 10 times more data than BERT, with a batch size eight times larger for half as many optimization steps, thus seeing four times as many sequences in pretraining compared to Devlin et al. (2019).

To help disentangle the importance of these factors from other modeling choices (e.g., the pretraining objective), we begin by training RoBERTa following the BERT$_{\text{LARGE}}$ architecture ($L = 24$, $H = 1024$, $A = 16$, 355M parameters). We pretrain for 100K steps over a comparable BOOKCORPUS plus WIKIPEDIA dataset as was used in Devlin et al. (2019). We pretrain our model using 1024 V100 GPUs, which takes approximately one day per 100K steps.

**Results** We present our results in Table 3. When controlling for training data, we observe that RoBERTa provides a large improvement over the originally reported BERT$_{\text{LARGE}}$ results, reaffirming the importance of the design choices we explored in Section 4.

Next, we combine this data with the three additional datasets described in Section 3.2. We train RoBERTa over the combined data with the same number of training steps as before (100K). In total, we pretrain over 160GB of text. We observe further improvements in performance across all downstream tasks, validating the importance of data size and diversity in pretraining.[9]

Finally, we pretrain RoBERTa for significantly longer, increasing the number of pretraining steps from 100K to 300K, and then further to 500K. We again observe significant gains in downstream task performance, and the 300K and 500K step models outperform XLNet$_{\text{LARGE}}$ across most tasks. We note that even our longest-trained model does not appear to overfit our data and would likely benefit from additional training.

## 5.1 GLUE RESULTS

For GLUE, we consider two finetuning settings. In the first setting (*single-task, dev*), we finetune RoBERTa separately for each of the GLUE tasks, using only the training data for the corresponding task. We consider a limited hyperparameter sweep with batch sizes $\in \{16, 32\}$ and learning rates $\in \{$1e-5, 2e-5, 3e-5$\}$, with a linear warmup for the first 6% of steps followed by a linear decay to 0. We finetune for 10 epochs with early stopping based on each task's dev set. The rest of the hyperparameters remain the same as during pretraining. In this setting, we report the median development set results for each task over five random initializations, without model ensembling.

In the second setting (*ensembles, test*), we compare RoBERTa to other approaches on the test set via the GLUE leaderboard. While many submissions to the GLUE leaderboard depend on multi-task

---

[9]A more careful analysis disentangling data size and diversity is needed, but is left to future work.

| | MNLI | QNLI | QQP | RTE | SST | MRPC | CoLA | STS | WNLI | Avg |
|---|---|---|---|---|---|---|---|---|---|---|
| *Single-task single models on dev* | | | | | | | | | | |
| BERT_LARGE | 86.6/- | 92.3 | 91.3 | 70.4 | 93.2 | 88.0 | 60.6 | 90.0 | - | - |
| XLNet_LARGE | 89.8/- | 93.9 | 91.8 | 83.8 | 95.6 | 89.2 | 63.6 | 91.8 | - | - |
| RoBERTa | **90.2/90.2** | **94.7** | **92.2** | **86.6** | **96.4** | **90.9** | **68.0** | **92.4** | **91.3** | - |
| *Ensembles on test (from leaderboard as of July 25, 2019)* | | | | | | | | | | |
| ALICE | 88.2/87.9 | 95.7 | **90.7** | 83.5 | 95.2 | 92.6 | **68.6** | 91.1 | 80.8 | 86.3 |
| MT-DNN | 87.9/87.4 | 96.0 | 89.9 | 86.3 | 96.5 | 92.7 | 68.4 | 91.1 | 89.0 | 87.6 |
| XLNet | 90.2/89.8 | 98.6 | 90.3 | 86.3 | **96.8** | **93.0** | 67.8 | 91.6 | **90.4** | 88.4 |
| RoBERTa | **90.8/90.2** | **98.9** | 90.2 | **88.2** | 96.7 | 92.3 | 67.8 | **92.2** | 89.0 | **88.5** |

Table 4: Results on GLUE. All results are based on a 24-layer architecture. BERT_LARGE and XLNet_LARGE results are from Devlin et al. (2019) and Yang et al. (2019), respectively. RoBERTa results on the dev set are a median over five runs. RoBERTa results on the test set are ensembles of *single-task* models. For RTE, STS and MRPC we finetune starting from the MNLI model.

| Model | SQuAD 1.1 | | SQuAD 2.0 | |
|---|---|---|---|---|
| | EM | F1 | EM | F1 |
| *Single models on dev, w/o data augmentation* | | | | |
| BERT_LARGE | 84.1 | 90.9 | 79.0 | 81.8 |
| XLNet_LARGE | **89.0** | 94.5 | 86.1 | 88.8 |
| RoBERTa | 88.9 | **94.6** | **86.5** | **89.4** |

| Model | SQuAD 2.0 | |
|---|---|---|
| | EM | F1 |
| *Single models on test (as of July 25, 2019)* | | |
| XLNet_LARGE | 86.3† | 89.1† |
| RoBERTa | 86.8 | 89.8 |
| XLNet + SG-Net Verifier | **87.0**† | **89.9**† |

Table 5: Results on SQuAD. † indicates results that depend on additional external training data. RoBERTa uses only the provided SQuAD data in both dev and test settings. BERT_LARGE and XLNet_LARGE results are from Devlin et al. (2019) and Yang et al. (2019), respectively.

finetuning, **our submission depends only on single-task finetuning**. For RTE, STS and MRPC we finetune starting from the MNLI single-task model, following Phang et al. (2018). We explore a slightly wider hyperparameter space, described in Appendix C, and ensemble between 5 and 7 models per task. Two of the GLUE tasks require task-specific finetuning approaches to achieve competitive leaderboard results; these approaches are described in Appendix B.

**Results** We present our results in Table 4. In the first setting (*single-task, dev*), RoBERTa achieves state-of-the-art results on all 9 of the GLUE task development sets. Crucially, RoBERTa uses the same masked language modeling pretraining objective and architecture as BERT_LARGE, yet consistently outperforms both BERT_LARGE and XLNet_LARGE. This raises questions about the relative importance of model architecture and pretraining objective, compared to more mundane details like dataset size and training time that we explore in this work. A more comprehensive comparison of the BERT and XLNet pretraining objectives is needed, but is left to future work.

In the second setting (*ensembles, test*), we submit RoBERTa to the GLUE leaderboard and achieve state-of-the-art results on 4 out of 9 tasks and the highest average score to date. Notably, RoBERTa does not depend on multi-task finetuning, and we expect future work may further improve these results by incorporating more sophisticated multi-task finetuning procedures.

## 5.2 SQUAD RESULTS

We adopt a much simpler approach for SQuAD compared to past work. While BERT (Devlin et al., 2019) and XLNet (Yang et al., 2019) augment their training data with additional QA datasets, we only finetune RoBERTa using the provided SQuAD training data. We also use a single learning rate for all layers, in contrast to the custom layer-wise learning rate scheduled used by Yang et al. (2019).

For SQuAD v1.1 we follow the same finetuning procedure as Devlin et al. (2019). For SQuAD v2.0, we additionally classify whether a given question is answerable; we train this classifier jointly with the span predictor by summing the classification and span loss terms.

| Model | Accuracy | Middle | High |
|---|---|---|---|
| *Single models on test (as of July 25, 2019)* | | | |
| BERT$_{\text{LARGE}}$ | 72.0 | 76.6 | 70.1 |
| XLNet$_{\text{LARGE}}$ | 81.7 | 85.4 | 80.2 |
| RoBERTa | **83.2** | **86.5** | **81.3** |

Table 6: Results on the RACE test set. BERT$_{\text{LARGE}}$ and XLNet$_{\text{LARGE}}$ results from Yang et al. (2019).

**Results**  We present our results in Table 5. On the SQuAD v1.1 development set, RoBERTa matches the state-of-the-art set by XLNet. On the SQuAD v2.0 development set, RoBERTa sets a new state-of-the-art, improving over XLNet by 0.4 points (EM) and 0.6 points (F1).

We also submit RoBERTa to the public SQuAD 2.0 leaderboard. Most of the top systems build upon either BERT (Devlin et al., 2019) or XLNet (Yang et al., 2019) and therefore rely on additional external training data. Our single RoBERTa model outperforms all but one of the single model submissions, and is the top scoring system among those that do not rely on additional external data.

### 5.3 RACE Results

In RACE, systems are provided with a passage of text, an associated question, and must classify which of four candidate answers is correct. We modify RoBERTa for this task by concatenating each candidate answer with the corresponding question and passage. We encode each of these four sequences and pass the resulting *[CLS]* representations through a fully-connected layer, which is used to predict the correct answer. We truncate question-answer pairs that are longer than 128 tokens and, if needed, the passage so that the total length is at most 512 tokens.

Results are presented in Table 6. RoBERTa achieves state-of-the-art accuracy across all settings.

## 6 Related Work

Pretraining methods have been designed with different training objectives, including language modeling (Dai & Le, 2015; Peters et al., 2018; Howard & Ruder, 2018), machine translation (McCann et al., 2017), and masked language modeling (Devlin et al., 2019; Lample & Conneau, 2019). Many recent papers have used a basic recipe of finetuning models for each end task (Howard & Ruder, 2018; Radford et al., 2018), and pretraining with some variant of a masked language model objective. However, newer methods have improved performance by multi-task fine tuning (Dong et al., 2019), incorporating entity embeddings (Sun et al., 2019), span prediction (Joshi et al., 2019), and multiple variants of autoregressive pretraining (Song et al., 2019; Chan et al., 2019; Yang et al., 2019). Performance is also typically improved by training bigger models on more data (Devlin et al., 2019; Baevski et al., 2019; Yang et al., 2019; Radford et al., 2019). Our goal was to replicate, simplify, and better tune the training of BERT, as a reference point for better understanding the relative performance of all of these methods.

## 7 Conclusion

We evaluate a number of design decisions when pretraining BERT models, demonstrating that performance can be substantially improved by training the model longer, with bigger batches over more data; removing the next sentence prediction objective; training on longer sequences; and dynamically changing the masking pattern applied to the training data. We additionally use a novel dataset, CC-News, and release our models and code for pretraining and finetuning at: `anonymous URL`.

Our improved pretraining procedure, which we call RoBERTa, achieves state-of-the-art results on GLUE, RACE, SQuAD, SuperGLUE and XNLI. These results illustrate the importance of these previously overlooked design decisions and suggest that BERT's pretraining objective remains competitive with recently proposed alternatives.

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

# A  STATIC VS. DYNAMIC MASKING

| Masking | SQuAD 2.0 | MNLI-m | SST-2 |
|---|---|---|---|
| reference | 76.3 | 84.3 | 92.8 |
| *Our reimplementation:* | | | |
| static | 78.3 | 84.3 | 92.5 |
| dynamic | 78.7 | 84.0 | 92.9 |

Table 7:   Comparison between the published BERT$_{\text{BASE}}$ results from Devlin et al. (2019) to our reimplementation with either static or dynamic masking. We report F1 for SQuAD and accuracy for MNLI-m and SST-2. Reported results are medians over 5 random initializations (seeds). Reference results are from Yang et al. (2019). We find that our reimplementation with static masking performs similar to the original BERT model, and dynamic masking is comparable or slightly better than static masking.

# B  TASK-SPECIFIC MODIFICATIONS FOR GLUE

Two of the GLUE tasks require task-specific finetuning approaches to achieve competitive leaderboard results:

**QNLI**   Recent submissions on the GLUE leaderboard adopt a pairwise ranking formulation for the QNLI task, in which candidate answers are mined from the training set and compared to one another, and a single (question, candidate) pair is classified as positive (Liu et al., 2019b;a; Yang et al., 2019). This formulation significantly simplifies the task, but is not directly comparable to BERT (Devlin et al., 2019). Following recent work, we adopt the ranking approach for our test submission, but for direct comparison with BERT all reported development set results are based on a pure classification approach.

**WNLI**   We found the provided NLI-format data to be challenging to work with. Instead we use the reformatted WNLI data from SuperGLUE (Wang et al., 2019a), which indicates the span of the query pronoun and referent. We then finetune RoBERTa using a variation of the approach from Kocijan et al. (2019). In particular, for a given input sentence, we first use spaCy (Honnibal & Montani, 2017) to extract additional candidate noun phrases from the sentence, and then finetune our model so that it assigns higher scores to positive referent phrases than for any of the generated negative candidate phrases.

In contrast to Kocijan et al. (2019), who finetune BERT using a margin ranking loss between (query, candidate) pairs, we instead use a single cross entropy loss term over the log-probabilities for the query and all mined candidates. This reduces the number of hyperparameters that need to be tuned and in practice produces more stable results on the development set. Our best model achieved 92.3% development set accuracy, compared to 90.2% accuracy for the margin loss approach.

One unfortunate consequence of our overall approach is that we can only make use of the positive training examples, which excludes over half of the provided training data.[10]

# C  FULL RESULTS ON GLUE

In Table 8 we present the full set of development set results for RoBERTa on all 9 GLUE datasets.[11] We present results for a LARGE configuration with 355M parameters that follows BERT$_{\text{LARGE}}$, as well as a BASE configuration with 125M parameters that follows BERT$_{\text{BASE}}$.

---

[10]While we only use the provided WNLI training data, our results could potentially be improved by augmenting this with additional pronoun disambiguation datasets.

[11]The GLUE datasets are: CoLA (Warstadt et al., 2018), Stanford Sentiment Treebank (SST) (Socher et al., 2013), Microsoft Research Paragraph Corpus (MRPC) (Dolan & Brockett, 2005), Semantic Textual Similarity Benchmark (STS) (Agirre et al., 2007), Quora Question Pairs (QQP) (Iyer et al., 2016), Multi-Genre NLI (MNLI) (Williams et al., 2018), Question NLI (QNLI) (Rajpurkar et al., 2016), Recognizing Textual Entail-

| | MNLI | QNLI | QQP | RTE | SST | MRPC | CoLA | STS |
|---|---|---|---|---|---|---|---|---|
| RoBERTa_BASE | | | | | | | | |
|   + all data + 500k steps | 87.6 | 92.8 | 91.9 | 78.7 | 94.8 | 90.2 | 63.6 | 91.2 |
| RoBERTa_LARGE | | | | | | | | |
|   with BOOKS + WIKI | 89.0 | 93.9 | 91.9 | 84.5 | 95.3 | 90.2 | 66.3 | 91.6 |
|   + additional data (§3.2) | 89.3 | 94.0 | 92.0 | 82.7 | 95.6 | **91.4** | 66.1 | 92.2 |
|   + pretrain longer 300k | 90.0 | 94.5 | **92.2** | 83.3 | 96.1 | 91.1 | 67.4 | 92.3 |
|   + pretrain longer 500k | **90.2** | **94.7** | **92.2** | **86.6** | **96.4** | 90.9 | **68.0** | **92.4** |

Table 8:   Development set results on GLUE tasks for various configurations of RoBERTa.  All results are a median over five runs.

## D   PRETRAINING HYPERPARAMETERS

| Hyperparam | RoBERTa_LARGE | RoBERTa_BASE |
|---|---|---|
| Number of Layers | 24 | 12 |
| Hidden size | 1024 | 768 |
| FFN inner hidden size | 4096 | 3072 |
| Attention heads | 16 | 12 |
| Attention head size | 64 | 64 |
| Dropout | 0.1 | 0.1 |
| Attention Dropout | 0.1 | 0.1 |
| Warmup Steps | 24k | 24k |
| Peak Learning Rate | 4e-4 | 6e-4 |
| Batch Size | 8k | 8k |
| Weight Decay | 0.01 | 0.01 |
| Max Steps | 500k | 500k |
| Learning Rate Decay | Linear | Linear |
| Adam $\epsilon$ | 1e-6 | 1e-6 |
| Adam $\beta_1$ | 0.9 | 0.9 |
| Adam $\beta_2$ | 0.98 | 0.98 |
| Gradient Clipping | 0.0 | 0.0 |

Table 9:  Hyperparameters for pretraining RoBERTa_LARGE and RoBERTa_BASE.

## E   FINETUNING HYPERPARAMETERS

| Hyperparam | RACE | SQuAD | GLUE | SuperGLUE |
|---|---|---|---|---|
| Learning Rate | 1e-5 | 1.5e-5 | {1e-5, 2e-5, 3e-5} | {1e-5, 2e-5, 3e-5} |
| Batch Size | 16 | 48 | {16, 32} | 32 |
| Weight Decay | 0.1 | 0.01 | 0.1 | 0.1 |
| Max Epochs | 4 | 2 | 10 | {10, 50} |
| Learning Rate Decay | Linear | Linear | Linear | Linear |
| Warmup ratio | 0.06 | 0.06 | 0.06 | 0.10 |

Table 10:  Hyperparameters for finetuning RoBERTa_LARGE on RACE, SQuAD and GLUE. We select the best hyperparameter values based on the median of 5 random seeds for each task.

ment (RTE) (Dagan et al., 2006; Bar Haim et al., 2006; Giampiccolo et al., 2007; Bentivogli et al., 2009) and Winograd NLI (WNLI) (Levesque et al., 2011).

## F    RESULTS ON SUPERGLUE

| | **BoolQ** | **CB** | **COPA** | **MultiRC** | **ReCoRD** | **RTE** | **WiC** | **WSC** | **Avg** |
|---|---|---|---|---|---|---|---|---|---|
| *Single-task single models on dev* | | | | | | | | | |
| BERT++ | 80.1 | 96.4/95.0 | 78.0 | 70.7/24.7 | 70.6/69.8 | 82.3 | 74.9 | 68.3 | 74.6 |
| RoBERTa | 86.9 | 98.2/- | 94.0 | 85.7/- | 89.5/89.0 | 86.6 | 75.6 | - | - |
| *Ensembles on test (from leaderboard as of August 12, 2019)* | | | | | | | | | |
| BERT | 77.4 | 75.7/83.6 | 70.6 | 70.0/24.1 | 72.0/71.3 | 71.7 | 69.6 | 64.4 | 69.0 |
| BERT++ | 79.0 | 84.8/90.4 | 73.8 | 70.0/24.1 | 72.0/71.3 | 79.0 | 69.6 | 64.4 | 71.5 |
| Outside Best | 80.4 | - | 84.4 | 70.4/24.5 | 74.8/73.0 | 82.7 | - | - | - |
| RoBERTa | **87.1** | **90.5/95.2** | **90.6** | **84.4/52.5** | **90.6/90.0** | **88.2** | **69.9** | **89.0** | **84.6** |
| Human (est.) | 89.0 | 95.8/98.9 | 100.0 | 81.8/51.9 | 91.7/91.3 | 93.6 | 80.0 | 100.0 | 89.8 |

Table 11: Results on SuperGLUE. All results are based on a 24-layer architecture. RoBERTa results on the development set are a median over five runs. RoBERTa results on the test set are ensembles of *single-task* models. Averages are obtained from the SuperGLUE leaderboard.

We also evaluate RoBERTa on the SuperGLUE benchmark (Wang et al., 2019a), which consists of 8 natural language understanding tasks.[12] We largely follow the same setup for SuperGLUE as we did for GLUE, with several task-specific modifications:

- **BoolQ** and **MultiRC**: we follow the same input format as the Wang et al. (2019a) baseline.
- **CB**: we finetune starting from the MNLI model, following Phang et al. (2018).
- **COPA**: we concatenate the premise and each alternative with `because` and `so` markers for *cause* and *effect* questions, respectively. This input format more closely matches the pretraining data format and provides better results in practice.
- **ReCoRD**: during training we adopt a pairwise ranking formulation with one negative and positive entity for each (passage, query) pair. At evaluation time, we pick the entity with the highest score for each question.
- **WiC**: we input the pair of sentences as normal. We then feed the concatenation of the representations of the two marked words and the *[CLS]* token to the classification layer.
- **RTE** and **WSC**: we reused our submission to the GLUE leaderboard.

In Table 11 we present RoBERTa results on the 8 SuperGLUE datasets. RoBERTa achieves state-of-the-art results on the development and test sets for BoolQ, CB, COPA, MultiRC and ReCoRD and the highest average score to date on the SuperGLUE leaderboard.

## G    RESULTS ON XNLI

| | en | fr | es | de | el | bg | ru | tr | ar | vi | th | zh | hi | sw | ur | Δ |
|---|---|---|---|---|---|---|---|---|---|---|---|---|---|---|---|---|
| *Machine translation baselines (TRANSLATE-TEST)* | | | | | | | | | | | | | | | | |
| XLM (MLM+TLM) | 85.0 | 79.0 | 79.5 | 78.1 | 77.8 | 77.6 | 75.5 | 73.7 | 73.7 | 70.8 | 70.4 | 73.6 | 69.0 | 64.7 | 65.1 | 74.2 |
| XLM-en | 88.8 | 81.4 | 82.3 | 80.1 | 80.3 | 80.9 | 76.2 | 76.0 | 75.4 | 72.0 | 71.9 | 75.6 | 70.0 | 65.8 | 65.8 | 76.2 |
| RoBERTa | **91.3** | **82.9** | **84.3** | **81.2** | **81.7** | **83.1** | **78.3** | **76.8** | **76.6** | **74.2** | **74.0** | **77.5** | **70.9** | **66.6** | **66.8** | **77.8** |

Table 12: Results on XNLI (Conneau et al., 2018) for RoBERTa$_{LARGE}$ in the TRANSLATE-TEST setting. We report macro-averaged accuracy (Δ) using the provided English translations of the XNLI test sets. RoBERTa achieves state of the art results on all 15 languages.

---

[12]The SuperGLUE datasets are: BoolQ (Clark et al., 2019), CommitmentBank (CB) (De Marneffe et al., 2019), Choice of Plausible Alternatives (COPA) (Roemmele et al., 2011), Multi-Sentence Reading Comprehension (MultiRC) (Khashabi et al., 2018), Reading Comprehension with Commonsense Reasoning (ReCoRD) (Zhang et al., 2018), Recognizing Textual Entailment (RTE) (Dagan et al., 2006; Bar Haim et al., 2006; Giampiccolo et al., 2007; Bentivogli et al., 2009), Words in Context (WiC) (Pilehvar & Camacho-Collados, 2019), the Winograd Schema Challenge (WSC) (Levesque et al., 2011), and Winogender Schema Diagnostics (Poliak et al., 2018; Rudinger et al., 2018).

