# OpenReview forum: "RoBERTa: A Robustly Optimized BERT Pretraining Approach"
_ICLR.cc/2020/Conference — Reject_

### Official Review · AnonReviewer3 · 2019-10-23
**Official Blind Review #3**

**Rating:** 6

**Review:**

This paper is a replication study of BERT for training large language models. Its main modification is simple: training longer with more data. Significantly improvements have been reported, and the work achieves on-par or higher accuracy over a large set of downstream tasks compared to XLNet, which is a state-of-the-art autoregressive language model.

Pros:
+ The paper incorporates robust optimization into BERT training with more data, and shows that together it significantly improves BERT's performance on downstream tasks.
+ The experimental results show that RoBERTa can significantly advance the baseline BERT model and achieve on-par or new state-of-the-art accuracy on a large range of downstream tasks.

Cons:
- While the replication study is well appreciated, the novelty contribution of the paper is marginally incremental as the model structure is largely unchanged from BERT. The other techniques applied also are somewhat trivial.
- Very little can be deduced from the experiments, as performance is often improved by training over more data.

Overall, I believe this paper comes at the right time and is addressing an interesting problem. The paper is well-organized and well-written. The contribution of the paper comes mostly from carefully taking into account several additional design choices and show that they could help train BERT with more data and can achieve SoTA performance on downstream tasks. Those modifications can be summarized as: (1) large-batch training with batch size 8k; (2) no mixed sequence length training only used 512 for the entire run; (3) no next sentence prediction; (4) dynamic masking instead of static; (5) larger byte-level BPE vocab (which increases BERT-large size by 20M parameters). Although they are interesting, a major concern is that it is difficult to find one thing that would have catapulted it over others to ensure publication.

Question:
Do you plan to release the datasets used for training in this work?

**Experience Assessment:**

I have read many papers in this area.

**Review Assessment: Checking Correctness Of Derivations And Theory:**

N/A

**Review Assessment: Checking Correctness Of Experiments:**

I carefully checked the experiments.

**Review Assessment: Thoroughness In Paper Reading:**

I read the paper at least twice and used my best judgement in assessing the paper.

---

> ### Author Response · Authors · 2019-11-13
> **Thank you for your review**
>
> Thank you for your review. Re: releasing of the data, we can’t release the raw text but we will provide resources for recreating the data. However, most of the datasets come from previous work and are available from the original sources (e.g., OpenWebText, Stories) or from third parties reproductions (BookCorpus, Wikipedia). In this work we also introduce the CC-NEWS dataset, which can be reconstructed using the news-please extractor: https://github.com/fhamborg/news-please/blob/master/newsplease/examples/commoncrawl.py.

---

### Official Review · AnonReviewer2 · 2019-10-23
**Official Blind Review #2**

**Rating:** 6

**Review:**

This paper presents a detailed replication study of the BERT pre-training model considering alternative design choices such as dynamic masking, removal of next sentence prediction loss, longer training time, larger batch sizes, additional training data, training on longer single document or cross-document sequences etc. to demonstrate their efficacy on several benchmark datasets and tasks by achieving the new state-of-the-art results. Overall, the paper is very well-written and the experimental setups are reasonable and thoroughly presented that would benefit the community for future research and exploration. However, I am not sure if the paper presents a case of adequate novelty in terms of ideas as many of them are rather obvious and the current state-of-the-art models could also improve considerably using similar experimental setups, which authors also acknowledged in footnote 2.

Other comments:

- Section 3.1: please clarify how exactly setting Beta2=0.98 would improve stability when training with large batch sizes.

- It's not clear what exactly was the motivation for proposing full-sentences and doc-sentences input formats. Please explain.

- Although the presented metrics show that the removal of NSP loss helps, however, no explanation was provided based on qualitative evaluation as to why this is the case. Some task-specific examples would have been nice to discuss the effects of NSP loss.

- Section 5: Please provide details on why you think the longest-trained model does not appear to overfit.

**Experience Assessment:**

I have published in this field for several years.

**Review Assessment: Checking Correctness Of Derivations And Theory:**

N/A

**Review Assessment: Checking Correctness Of Experiments:**

I carefully checked the experiments.

**Review Assessment: Thoroughness In Paper Reading:**

I read the paper thoroughly.

---

> ### Author Response · Authors · 2019-11-13
> **Thank you for your review**
>
> Thank you for your review. Regarding our use of a smaller beta2, good catch. This is primarily to improve stability when training with *larger learning rates*, but is not directly related to batch size. We will correct and clarify this in an updated version of the paper.
>
> Re: “full-sentences” and “doc-sentences” input formats, the primary motivation was to remove the NSP loss. The tradeoffs between these two input formats relate to efficiency and potential noise from input segments that span multiple documents. The “full-sentences” format samples sentences contiguously from the corpus, but inputs may cross document boundaries and thus the inputs are slightly noisier — a similar noise occurs with NSP when sampling negative samples from distinct documents. “doc-sentences” inputs can only come from a single document and thus does not contain cross-document noise. However, to maintain efficiency the doc-sentences format requires us to use a dynamic batch size that makes it more difficult to compare to other work. We will clarify this and add additional discussion around these points.
>
> Re: NSP loss, we’re not able to observe any clear qualitative differences, and even quantitative differences are very small. Indeed, one of our contributions is showing that the NSP loss does not have a meaningful effect on downstream task performance, thus we remove it.
>
> Re: overfitting, we observed that the validation loss continued to decrease even for our longest-trained model, suggesting that it may benefit from even longer training. We will clarify this.

---

### Official Review · AnonReviewer1 · 2019-10-25
**Official Blind Review #1**

**Rating:** 6

**Review:**

This paper presents a replication study of BERT pretraining and carefully measures the impact of many key hyperparameters and training data size. It shows that BERT was significantly undertrained and propose an improved training recipe called RoBERTa. The key ideas are: (i) training longer with bigger batches over more data, (ii) removing NSP, (iii) training over long sequences, and (iv) dynamically changing the masking pattern. The proposed RoBERTa achieves/matches state-of-the-art performance on many standard NLU downstream tasks.

The in-depth experimental analysis of the BERT pretraining process in this paper answers many open questions (e.g., the usefulness of NSP objective) and also provide some guidance in how to effectively tweak the performance of pretrained model (e.g., large batch size). It also further demonstrates that the BERT model, once fully tuned, could achieve SOTA/competitive performance compared to the recent new models (e.g., XLNet). The main weakness of the paper is that it is mainly based on further tuning the existing BERT model and lacks novel contribution in model architecture. However, the BERT analysis results provided in this paper should also be valuable to the community.

Questions & Comments:
•	It is stated that the performance is sensitive to epsilon in AdamW. This reminds us of the sensitivity of BERT pretraining to the optimizers. Since one of the main contributions of this paper is the analysis of the BERT pretraining process, more experimental analysis on the optimizer should also be included.
•	It is stated that (page 7) the submission to GLUE leaderboard uses only single-task finetuning. Is there any special reason for restraining it to single-task finetuning if earlier results demonstrates multi-task finetuning is better? Of course, it is valuable to see the great performance achieved by single-task finetuning for RoBERTa. But there should be no reason that it is restricted to be so. An additional experimental results with multi-task finetuning should also be added.


**Experience Assessment:**

I have published in this field for several years.

**Review Assessment: Checking Correctness Of Derivations And Theory:**

I carefully checked the derivations and theory.

**Review Assessment: Checking Correctness Of Experiments:**

I carefully checked the experiments.

**Review Assessment: Thoroughness In Paper Reading:**

I read the paper thoroughly.

---

> ### Author Response · Authors · 2019-11-13
> **Thank you for your review**
>
> Thank you for your review. Regarding the sensitivity of pretraining to the AdamW epsilon term, we note that our setting (1e-6) matches that of the original BERT paper. While most deep learning platforms use a default value of eps=1e-8, we found that BERT would often fail to train with eps=1e-8 or 1e-7, especially for the larger models. An epsilon of 1e-6 was the smallest value that worked consistently, so we added a remark about sensitivity to this hyperparameter in case it was helpful for others. We will clarify this in an updated version of the paper.
>
> Regarding multi-task finetuning, the focus of our work was primarily to evaluate the importance of various design decisions in the BERT pretraining phase. While multi-task finetuning is often effective (and is itself an active area of research), we feel that it goes beyond the scope of our paper which focuses primarily on pretraining. We agree it is worth studying in future work.

---

### Author Response · Authors · 2019-11-13
**Thank you for your reviews and helpful comments**

We thank the three reviewers for their helpful comments. All reviewers note that the paper lacks novelty in model architecture. We agree but would argue there is some novelty in the overall training scheme. However, model/training novelty isn’t the main contribution; our goal is to provide a better understanding of which factors are the most important for effective model pretraining, which we hope will be of very broad and general interest. All three reviewers agree that the paper will provide benefit to the community and will guide future research. In fact, several follow-ups have already been made public.

---

### Decision · Program_Chairs · 2019-12-19

**Decision:**

Reject

**Comment:**

This paper conducts an extensive study of training BERT and shows that its performance can be improved significantly by choosing a better training setup (e.g., hyperparameters, objective functions). I think this paper clearly offers a better understanding of the importance of tuning a language model to get the best performance on downstream tasks. However, most of the findings are obvious (careful tuning helps, more data helps). I think the novelty and technical contributions are rather limited for a conference such as ICLR. These concerns are also shared by all the reviewers. The review scores are borderline, so I recommend to reject the paper.